# Partial Purification and Biochemical Evaluation of Protease Fraction (MA-1) from *Mycoleptodonoides aitchisonii* and Its Fibrinolytic Effect

**DOI:** 10.3390/antiox12081558

**Published:** 2023-08-04

**Authors:** Sung-Ho Lee, Seung-Yub Song, Jun-Hui Choi, Seung Kim, Hyo-Jeong Lee, Jin Woo Park, Dae-Hun Park, Chun-Sik Bae, Seung-Sik Cho

**Affiliations:** 1Department of Pharmacy, College of Pharmacy, Mokpo National University, Muan 58554, Republic of Korea; tjdgh0730@naver.com (S.-H.L.); tgb1007@naver.com (S.-Y.S.); jwpark@mnu.ac.kr (J.W.P.); 2Department of Biomedicine, Health & Life Convergence Sciences, BK21 Four, Biomedical and Healthcare Research Institute, Mokpo National University, Mokpo 58554, Republic of Korea; 3Department of Food Science and Biotechnology, Gwangju University, Gwangju 61743, Republic of Korea; sekai0572@naver.com (J.-H.C.); seungk@gwangju.ac.kr (S.K.); jhshkl@gwangju.ac.kr (H.-J.L.); 4College of Oriental Medicine, Dongshin University, Naju-si 58245, Republic of Korea; dhj1221@hanmail.net; 5College of Veterinary Medicine, Chonnam National University, 77 Yongbong-ro, Buk-gu, Gwangju 61186, Republic of Korea; csbae210@chonnam.ac.kr

**Keywords:** *Mycoleptodonoides aitchisonii*, fibrinolytic enzyme, acidic tolerance antithrombotic, thromboembolism

## Abstract

The antioxidative proteolytic fraction, MA-1, was partially purified from *Mycoleptodonoides aitchisonii*. MA-1 was purified to homogeneity using a two-step procedure, which resulted in an 89-fold increase in specific activity and 42.5% recovery. SDS-PAGE revealed two proteins with a molecular weight of 48 kDa. The zymography results revealed proteolytic activity based on the MA-1 band. MA-1 was found to be stable in the presence of Na^+^, Ca^2+^, Fe^3+^, K^+^, and Mg^2+^. MA-1 was also stable in methanol, ethanol, and acetone, and its enzyme activity increased by 15% in SDS. MA-1 was inhibited by ethylenediaminetetra-acetic acid or ethylene glycol tetraacetic acid and exerted the highest specificity for the substrate, MeO-Suc-Arg-Pro-Tyr-pNA, for chymotrypsin. Accordingly, MA-1 belongs to the family of chymotrypsin-like metalloproteins. The optimum temperature was 40 °C and stability was stable in the range of 20 to 35 °C. The optimum pH and stability were pH 5.5 and pH 4–11. MA-1 exhibited stronger fibrinolytic activity than plasmin. MA-1 hydrolyzed the Aα, Bβ, and γ chains of fibrinogen within 2 h. MA-1 exhibited an antithrombotic effect in animal models. MA-1 was devoid of hemorrhagic activity at a dose of 80,000 U/kg. Overall, our results show that *M. aitchisonii* produces an acid-tolerant and antioxidative chymotrypsin-like fibrinolytic enzyme, and *M. aitchisonii* containing MA-1 could be a beneficial functional material for the prevention of cardiovascular diseases and possible complications.

## 1. Introduction

*Mycoleptodonoides aitchisonii* is an edible mushroom belonging to the family, Climacodontaceae, that is mainly found in India and Japan [1]. This mushroom has a cap that is approximately 3–8 cm in diameter, its color ranges from white to yellow, its surface is smooth, and its stem is short. *M. aitchisonii* grows on the trunks of dead beech trees. Notably, *M. aitchisonii* is an edible mushroom that is commonly used as a natural health product [2].

Recently, Republic of Korea has succeeded in the mass production of *M. aitchisonii* through artificial cultivation [3]. Recently, studies on the biological efficacy of *M. aitchisonii*, the toxicity of single and repeated administration of hot water extracts [4], the anti-asthma [5] and immune-stimulating effects of polysaccharides [6], and the anti-obesity effects [7] of *M. aitchisonii* dried powder, have been conducted. The methanol extract of *M. aitchisonii* has been reported to have an anti-inflammatory effect [8]. Further, the hot water extract has been found to lower blood pressure in animal models [9]. 

The polysaccharide fractions of *M. aitchisonii* have been reported to increase nerve growth factor (NGF) and catecholamine levels in rat brains [10]. In addition, phenylpentane compounds from *M. aitchisonii* mycelia enhance dopamine release in rat brain striatal slices [11]. In particular, inducing transient bilateral carotid artery occlusion in rats and the oral administration of *M. aitchisonii* or a polysaccharide extract can reduce ischemic damage by affecting monoamine metabolism in the cerebral cortex [12].

The bioactive compounds in mushroom extract include 5-hydroxy-4-(1-hydroxyethyl)-3-methylfuran-2(5H)-one and (3R,4R)-4-((R)-), which are endoplasmic reticulum stress inhibitors. 1-Hydroxyethyl-3-methyldihydrofuran-2(3H)-one, (3S,4R)-5-Phenylpentane-1,3,4-triol [13], and phenylpentane-based volatile substances, such as 1-phenyl-3-pentanol and 1-phenyl-3-pentanone, are known [11].

Although polysaccharides and some compounds of *M. aitchisonii* have been identified and their physiological activities are known, no studies have been conducted on biologically active catalysts. Therefore, in the present study, *M. aitchisonii* was treated with cold water to obtain crude proteins and polysaccharides, and the protease fractions (MA-1) were obtained via gel permeation column chromatography. Partially purified MA-1 was confirmed to affect optimum temperature, pH, stability, metal ions, enzyme inhibitors, and surfactants. In addition, MA-1 was confirmed to exhibit fibrino(geno)lytic activity and act as a chymotrypsin-like metalloprotease. The characteristics of MA-1 were revealed through an evaluation of its blood clot-degrading mechanism and characteristics in vivo.

## 2. Materials and Methods

### 2.1. Materials

Bovine fibrinogen, human fibrinogen, human fibrin, bovine thrombin, human thrombin, plasmin, azocasein, tris (hydroxymethyl) aminomethane (Tris), tetramethyl-ethylenediamine (TEMED), sodium dodecyl sulfate-polyacrylamide gel electrophoresis, phenylmethanesulfonyl fluoride (PMSF), ethylenediaminetetra-acetic acid (EDTA), and ethylene glycol tetraacetic acid (EGTA) were purchased from Sigma-Aldrich Co. (Burlington, NJ, USA). The protein markers used for SDS-PAGE were purchased from Fermentas Co. (Burlington, MA, USA). Sephadex resin was purchased from GE Healthcare Co. (Boston, MA, USA). The substrates for the amidolytic assay were purchased from Diapharma (West Chester, OH, USA). Other reagents were of special grade and were purchased commercially.

### 2.2. Rodents

ICR mice (25 g, 6 weeks of age, male) and Sprague-Dawley (SD) rats (200–300 g, male) were used in the tests. Four mice and four rats were housed in each cage. Mice were reared at 22 ± 2 °C, with a 12 h light/dark cycle. Feed and water were provided ad libitum. The environment was created to minimize the stress on the animals. All experimental procedures were performed under the guidance of the National Institutes of Health Guide for the Care and Use of Laboratory Animals [13]. It was approved by the Ethics Committee of Chonnam National University (CNU-IACUC-YB-2019-47).

### 2.3. Preparation of the Crude Extract and Purification of MA-1

*M. aitchisonii* was supported by the Jeollanamdo Wando Arboretum (Wando, Republic of Korea) and was prepared based on our previous report [5]. The crude protein extract was prepared using 250 g of mycelium powder in 1000 mL filtered sterile water and agitated for 24 h at 10 °C. Insoluble material was removed via filtration and the water extract was freeze-dried. The purification steps were performed under 10 °C. The crude was fractionated via gel permeation chromatography (GPC) using a Sephadex G-100 column (1 × 100 cm) equilibrated with 10 mM Tris-HCl (pH 7.0) at a flow rate of 2 mL/fraction. Loading volume was 1 mL and the protein amount was 25 mg. The protease activity of the active parts was determined using an azocasein method (at 660 nm) as described below. The protein concentration was determined using the Bradford method (at 595 nm) [14]. Fractions exerting protease activity were concentrated using a Vivaspin (Millipore, Burlington, MA, USA) and used as purified enzymes (MA-1) for further analysis.

### 2.4. Protease Assay

Protease activity was determined using a previously described method with minor modifications [15]. The mixture containing 50 μL azocasein (2% solution in 10 mM Tris-HCl buffer pH 8.0) and samples (100 μL) were incubated at 40 °C for 60 min and then stopped with 50 μL of trichloroacetic acid (10%). After centrifugation at 13,000× *g* for 10 min, 150 μL of the supernatant was mixed with 300 μL Folin–Ciocalteu’s phenol reagent (0.33 M) and 450 μL Na_2_CO_3_ solution (10%, *w*/*v*), and absorbance was measured at 660 nm. Enzyme activity was converted to a unit that has the ability to produce 1 μg of tyrosine per minute.

### 2.5. Biochemical Properties of MA-1

Determination of the optimal reaction temperature was performed by comparing enzyme activity at various temperatures under the condition of pH 7.0 (20 to 60 °C). For temperature stability, the remaining activity was measured after leaving the enzyme in the range of 20 to 55 °C degrees for 2 h. The optimal pH for enzyme activity was determined by measuring the residual enzyme activity in various pH conditions (3~11). The pH stability was evaluated by measuring the remaining activity after standing at 35 °C for 2 h under various pH conditions. The effects of metal ions and enzyme inhibitors on protease activity were also assessed using NaCl_2_, MgCl_2_, CaCl_2_, FeCl_3_, KCl, EDTA, EGTA, and PMSF (1 mM). The effects of organic solvents and surfactants on protease activity were also evaluated with methanol, ethanol, acetone, sodium dodecyl sulfate, triton ×100, and tween 20 (10%, *v*/*v*). The enzyme was mixed with 1 mM of different metal ions and 10% solvents and detergents. Finally, the protease activity was measured using azocasein.

The amidolytic activity based on several chromogenic substrates was determined by mixing 25 μL MA-1 (5 mg/mL) and 100 μL tris buffer (pH 7.5; 10 mM) with 75 μL of substrates (4 mM), such as H-D-Phe-Pip-Arg-pNA (S-2238; for thrombin); H-D-Val-Leu-Lys-pNA (S-2251; for plasmin); H-D-Ile-Pro-Arg-pNA (S-2288; for t-PA); MeO-Suc-Arg-Pro-Tyr-pNA (DPG-586; for chymotrypsin); and Pyro-Glu-Gly-Arg-pNA (DPG-444; for u-PA) in 20 mM Tris-HCl (pH 7.5) in a 1.5 mL tube. The mixtures were incubated at 37 °C for 10 min and the amount of *p*-nitroaniline (or the substrate) released was determined at 405 nm [16].

### 2.6. Antioxidant Assay

DPPH radical scavenging assay was determined according to Rahman’s protocol. Briefly, the sample was added to the DPPH solution (0.4 mM), mixed for 10 min, and measured at 517 nm [17].

The total phenolic content was determined by the Folin reaction [18]. The samples were mixed with Na_2_CO_3_ solution and Folin–Ciocalteu phenol reagent for 10 min. The mixture was measured at 750 nm. Gallic acid was used as standard. Results were expressed as milligrams of gallic acid equivalents per gram of sample. 

The reducing power was evaluated to test the antioxidant properties of the sample. The extract (0.1 mL) was mixed with 0.2 M sodium phosphate buffer (0.5 mL, pH 6.5) and K_3_[Fe(CN)_6_] (0.5 mL, 1% *w*/*v*), followed by incubation at 50 °C for 20 min, and the reaction was stopped by adding TCA (0.5 mL, 10%). After centrifugation, the supernatant was mixed with distilled water (0.5 mL) and FeCl_3_ (0.1 mL, 0.1% *w*/*v*), and the absorbance of the mixture was measured at 700 nm. The reducing powers were expressed as vitamin C equivalents [19].

### 2.7. Fibrinolytic and Fibrinogenolytic Activity Assay

The fibrinolytic activity of MA-1 and the positive control (plasmin from human plasma) were evaluated. MA-1 (1 U) and the control (5 U) were prepared on fibrin plates for the fibrinolytic activity assay. The fibrinolytic activity was analyzed using fibrin plates according to the method described by [20]. The fibrin plates were prepared by pouring the solution comprising 4.5 mg/mL fibrinogen in 10 mM Tris-HCl buffer (pH 7.0), 1.2% agarose, and 0.45 U/mL thrombin into a Petri dish. The solution was incubated for 30 min at room temperature for fibrin clot formation. MA-1 and plasmin (1 and 5 U) were loaded onto a hole (3.5 mm in diameter) of the plate, which was then incubated at 37 °C for 24 h. The fibrinolytic activity was estimated by measuring the diameter of the clear zone around the well. 

Quantitative analyses of fibrinogenolytic activity were conducted according to a previously published method, with slight modifications [20]. Briefly, 200 μL of fibrinogen (10 mg/mL) in 10 mM Tris-HCl buffer (pH 7.5) comprising 0.15 M NaCl was incubated at 37 °C with 60 μL MA-1 (0.6 U) for various durations. The incubated sample (10 µL) was withdrawn at the indicated time intervals, then boiled for 5 min to stop the reaction. The resulting degradation products were then analyzed via SDS-PAGE (12% separating gel).

### 2.8. Ex Vivo Coagulation Assay

Ex vivo coagulation assays were performed according to the method from Broersma et al. [21]. Six experimental groups (ten animals per group) were used: group 1 (saline treatment), groups 2–3 (40,000 and 80,000 U/kg nattokinase), and groups 4–6 (20,000, 40,000, and 80,000 U/kg MA-1). Nattokinase, a well-known thrombolytic enzyme, was selected as the control and purified according to a previously reported method [22]. The test samples (saline, nattokinase, and MA-1) were administered intravenously into the tail vein 30 min before sampling. Blood was collected after treatment on sodium citrate solution (3.8%), and plasma was obtained via centrifugation of the blood samples at 2000 g for 10 min. Activated partial thromboplastin time (APTT) and prothrombin time (PT) was performed according to the instructions provided by the manufacturer (Fisher Diagnostics, Middletown, CT, USA).

### 2.9. Acute Thromboembolism Animal Model

The collagen and epinephrine-induced thromboembolism model was established in ICR mice according to the method described by Dimmino et al. [23]. Five groups (ten animals/group) were established: group 1 (vehicle), group 2 (collagen + epinephrine), group 3 (nattokinase, 80,000 U/kg + collagen + epinephrine), and groups 4 and 5 (40,000, and 80,000 U/kg MA-1 + a mixture of collagen and epinephrine). One hour before the thrombotic challenge, mice were injected intraperitoneally with nattokinase and MA-1. Thereafter, mice were rapidly injected with 0.1 mL of a mixture of collagen and epinephrine via the tail vein. Mortality or paralysis was recorded for 15 min, and all surviving mice were killed immediately after the experiment.

### 2.10. FeCl_3_-Induced Carotid Arterial Thrombus Model

FeCl_3_-induced arterial thrombosis was established as described previously, with some modifications [24]. Briefly, male SD rats were anesthetized with an intraperitoneal injection of 50 mg/kg pentobarbital. The tested samples (saline, nattokinase (20,000 U/kg) and MA-1 (10,000, 20,000 U/kg)) were administered intravenously 1 min before FeCl_3_ injury. Then, the right carotid artery segment was exposed by blunt dissection. Two sheets of filter paper (6 mm × 6 mm) were treated with a 4% FeCl_3_ solution and then placed on opposite sides of the carotid artery in contact with the lateral surface of the vessel for 3 min. After 30 min, the filter paper was removed, and the mice were sacrificed and perfused with 4% paraformaldehyde through cardiac puncture. The secured carotid artery was fixed with 4% paraformaldehyde for 12 h at 4 °C. The obtained carotid arteries were made into frozen sections and H&E staining was performed to evaluate the antithrombotic effect of MA-1. Quantitative evaluation of thrombi was performed using ImageJ 1.46 b image analysis software.

### 2.11. Hemostasis Assessment

Tail bleeding time was measured according to Kim et al.’s method [24]. Mice were anesthetized and the t-PA (5000 U/kg), nattokinase (80,000 U/kg), and MA-1 (80,000 U/kg) were injected intravenously via the tail veins of mice. After 30 min, the mice were placed on a hot block (37 °C) and the end of the tail was cut, and then immersed in a tube containing saline. The bleeding time was recorded for 30 min. 

### 2.12. Statistical Analysis

Data are expressed as mean ± SD. Statistical significance was calculated using a post hoc test for multiple group comparisons. Differences with *p* values less than 0.01 and 0.05 were considered statistically significant.

## 3. Results and Discussion

### 3.1. Purification of the Protease Fraction, MA-1

We partially purified the protease fraction, MA-1, from *M. aitchisonii* using the steps listed in Table 1. Two proteins were obtained via gel permeation chromatography (GPC, Sephadex G-100) and were found to have similar molecular weights. Approximately 25 mg of crude protein was obtained after cold water extraction. The crude enzyme was subjected to gel permeation chromatography (GPC) on a Sephadex G-100 column to collect fractions that exhibited proteolytic activity (Figure 1B). Finally, the partially purified enzyme (MA-1) was subjected to SDS-PAGE. MA-1 appeared as two bands on the PAGE (Figure 1A). The final yield of the active fraction from 25 mg of crude protein was 0.12 mg (Table 1). The protease activity of the purified MA-1 was verified by gelatin zymography (Figure 1A).

### 3.2. MA-1 Showed Fibrino(geno)lytic Activity

The fibrinolytic effect of MA-1 on fibrin was analyzed using the fibrin plate method. In Figure 1C, the size of the fibrin cleavage zone after treatment with 1 U MA-1 was similar to that after treatment with 5 U plasmin, suggesting that the fibrinolytic efficiency of MA-1 was greater than that of plasmin. The clear zone owing to MA-1 was 14 mm/U, which was approximately 5-fold higher than that produced by plasmin (17 mm/5 FU) after 24 h of incubation at 37 °C. Additionally, to elucidate the mode of the fibrinogenolytic effect of MA-1, Aα, Bβ, and γ chains were analyzed using SDS-PAGE. As shown in Figure 1D, the Aα-chains were completely cleaved by the enzyme within 5 min and the Bβ-chains were almost completely cleaved within 120 min. Moreover, the enzyme hydrolyzed the γ-chains within 60 min of incubation.

### 3.3. Biochemical Properties of MA-1

The optimal reaction temperature for MA-1 was determined at various reaction temperatures. The thermal stability of the enzyme was examined via incubation at different temperatures for 2 h. As shown in Figure 2A, MA-1 was active at temperatures between 35 and 45 °C and exhibited maximal activity at 40 °C. The activity of MA-1 was stable at 20–35 °C (Figure 2B). The optimal reaction pH of MA-1 was determined using buffers with various pH values. As shown in Figure 2C, MA-1 was active at pH values of 5.5, 6, and 9. MA-1 was stable in the pH range of 4 to 11 at 4 °C for 2 h. The effect of pH on enzyme activity was also examined. As shown in Figure 2D, MA-1 exhibited a wide stability range from pH 4 to 11.

The effects of several metal ions, protease inhibitors, organic solvents, and surfactants are summarized in Table 2. The enzyme was inhibited by 1 mM ethylenediaminetetraacetic acid (EDTA) and ethyleneglycol tetraacetic acid (EGTA), which are inhibitors of metalloproteases. However, MA-1 was not inhibited by PMSF, a protease inhibitor that reacts with proteases like chymotrypsin, trypsin, and papain. To further investigate the characteristics of MA-1, its amidolytic activity was evaluated using various chromogenic substrates (Figure 3). The enzyme exhibited the highest specificity for the MeO-Suc-Arg-Pro-Tyr-pNA substrate for chymotrypsin, whereas MA-1 exhibited weak activity for the substrates t-PA, thrombin, plasmin, and u-PA. These findings suggest that MA-1 has hydrolytic characteristics similar to those of serine proteases or serine metalloproteases.

The effects of metal ions, solvents, and surfactants on enzyme activity were examined. MA-1 was incubated with 1.0 mM of metal ions, 10% of solvents, and 10% of surfactants and residual enzyme activity was measured. MA-1 was not inhibited by the metal ions. In the various solvent tests, 10% (*v*/*v*) ethanol, methanol, and acetone did not decrease the enzyme activity; however, Triton X-100 (approximately 34.6% reduction), glycerol (approximately 4.7% reduction), and Tween 20 (approximately 17.6% reduction) decreased the activity. SDS increased enzyme activity (approximately 16.2%) (Table 2). The DPPH radical scavenging ability of MA-1 was 82.83% at a concentration of 5 mg/mL, and ascorbic acid, a control, showed 85.92% at a concentration of 25 μg/mL. The reductive activity of MA-1 expressed as vitamin C equivalents was 20.16 ± 1.28 μg/100 μg protein. 

### 3.4. MA-1 Showed Excellent Anti-Coagulant Effect in an Ex Vivo Coagulation Model

By measuring the APTT and PT, we determined the coagulant factors (extrinsic and intrinsic) of MA-1 that exhibited anticoagulant effects. The ex vivo test was conducted by dividing mice into the control and experimental groups (groups treated with MA-1). The effects of MA-1 (20,000–80,000 U/kg) on the APTT and PT were evaluated ex vivo. MA-1 delayed the APTT in a dose-dependent manner. At a dose of 80,000 U/kg, MA-1 prolonged the APTT by 3.38 fold (120.8 ± 4.8 s) compared with the control (35.5 ± 1.8 s) (Table 3). However, MA-1 did not affect the PT compared with the control (12.3 ± 0.4 s). Our findings indicate that MA-1 may possess anticoagulant properties by prolonging the common and/or intrinsic coagulation pathways.

### 3.5. MA-1 Showed Effects of Inhibiting Clot Formation and Delaying Bleeding in Animal Models of Thrombosis and Hemostasis

In the present study, we experimented with the acute thromboembolism model and a FeCl_3_-induced arterial thrombosis model to evaluate the antithrombotic activity of MA-1. Collagen with epinephrine treatment caused 100% death or paralysis in the vehicle-treated group. Nattokinase (80,000 U/kg/day) was used as a positive control and exhibited a 50% protection rate against death and paralysis and the MA-1-treated group displayed 30% and 40% protection rates when administered 40,000 U/kg/day and 80,000 U/kg/day.

The FeCl_3_-induced arterial thrombosis model was employed as another animal model in this study. The percentage occlusion of the right carotid artery was determined to investigate the antithrombotic activity of MA-1 and nattokinase as positive controls. Figure 4 shows the qualitative and quantitative percentages of occlusion of the right carotid. As shown in Figure 4B, the FeCl_3_-induced vehicle-treated group displayed perfect occlusion compared with the control group (Figure 4A). As shown in Figure 4C, the positive control had an occlusion inhibition percentage of 24.1 ± 2.95% (*p* < 0.01) due to nattokinase, while the MA-1-treated group had percentages of 10.29 ± 1.42% and 16.62 ± 1.15% (*p* < 0.01) compared to the FeCl_3_ + saline-treated group (Figure 3D,E). Based on our findings, the antithrombotic activity of MA-1 was confirmed in the FeCl_3_-induced arterial thrombosis model and compared with that of nattokinase. Although the MA-1 exhibited antithrombotic activity compared with the vehicle, its occlusion inhibition was lower than that of nattokinase. 

To assess the effect of MA-1 on hemostasis, saline, nattokinase (80,000 U/kg), MA-1 (80,000 U/kg), and t-PA (5000 U/kg) were administered and the tail bleeding time was measured after cutting off the tip. Hemostasis was achieved at approximately 410, 1620, 570, and 660 s with saline, t-PA, nattokinase, and MA-1, respectively (Table 4). MA-1 displayed a similar or slightly longer hemostasis time than nattokinase, increasing the hemostasis time by 1.6 times compared to that of the control group treated with saline. However, compared to t-PA, which increased the hemostasis time by approximately 4 times compared to the control, the hemostasis time of the experimental group treated with MA-1 was confirmed to be relatively short.

## 4. Discussion

Since the report of the anti-inflammatory efficacy of the *M. aitchisonii* extract in the 1990s, research on its biological activity and active ingredients has been steadily progressing. In the last decade, toxicity and bioactivity evaluations of the functional materials of *M. aitchisonii* extracts have been conducted. Hot water extracts have been reported to be closely associated with immune regulation, improvement of brain function, antihypertensive effects, and improvement of ischemic damage [1,5,6,7,12,13]. Recently, the hot water extracts of artificially grown *M. aitchisonii* have been reported to contain a high polysaccharide content of 30% or more. Therefore, the need to develop *M. aitchisonii* as a functional health material for ingestion as a dry powder or aqueous extract has increased. To date, no reports have described the purification and functionality of functional catalysts in the water-soluble fractions of *M. aitchisonii*. Therefore, we partially purified the proteolytic enzyme obtained from *M. aitchisonii* at a low temperature to investigate its functionality.

Intravascular thrombosis, the accumulation of intravascular fibrin, is caused by the accumulation of fibrin in blood vessels. Thrombosis causes cardiovascular disease and myocardial infarction, which is statistically known to cause more than 17 million deaths annually [25]. In general, injections are commonly used for the treatment of thrombosis, and research on fibrinolytic agents that can be administered orally is ongoing [26,27]. Thrombosis is closely associated with fibrin clot formation. Fibrin is degraded by plasmin (EC 3.4.21.7), and inactive plasminogen is converted into active plasmin by the tissue-type plasminogen activator (tPA) (EC 3.4.21.73). Urokinase (EC 3.4.21.73), streptokinase (EC 3.4.99.22), and tissue-type plasminogen activator (tPA) are typical fibrinolytic agents [28].

The fibrinolytic agents used for medical purposes have a strong clot-dissolving action, but have a limited route of administration and side effects, such as bleeding. Therefore, studies are being actively conducted to reduce the side effects and develop materials that can be administered orally. Attempts have been made to develop fibrinolytic enzymes and materials containing fibrinolytic enzymes via daily oral intake as food sources to prevent intravascular and/or cardiovascular diseases.

Thrombolytic enzymes have been found in microorganisms [29], fungi [30], marine organisms [31], and leeches [32]. In particular, Bacillus Nattokinase produced by bacteria in the genus can be administered orally and is known to have excellent fibrinolytic activity [33]. Therefore, in this study, fibrinolytic enzymes were compared with nattokinase and MA-1 in vivo to assess thrombolysis.

We prepared a water extract of dried *M. aitchisonii* at a low temperature (10 °C) and freeze-dried this extract. A highly active proteolytic enzyme fraction (MA-1) was then obtained through one-step column purification. Using sodium dodecyl sulfate-polyacrylamide gel electrophoresis, two protein mixtures of similar sizes (approximately 48 kDa) were identified. Fibrinolytic enzymes with a size similar to that of MA-1 can be found in some microorganisms. *Enterobacterium serratia* E–15 is reported to induce fibrinolysis and can distinguish and dissolve dead and damaged tissues without harming cells. Serralysin, a proteolytic enzyme of less than 50 kDa [34,35], *Shewanella* sp. IND20, and *Psuedoalteromonas* sp. IND11 produce thrombolytic enzymes with molecular weights of 44 kDa and 64 kDa, and exhibit direct coagulolytic activity and proenzyme plasminogen (PLG) activating ability [36,37].

MA-1 exhibited optimal proteolytic activity at pH 5.5 and 40 °C, and high activity even at around pH 9, which might be due to the different physicochemical characteristics of the two proteins in MA-1. MA-1 was stable when left for 2 h at pH 5, and 20–35 °C. When MA-1 was left for 2 h, it exhibited more than 90% residual activity at pH 3 to 11. These results suggest that MA-1 is composed of two types of proteins and the MA-1 fraction must be further subdivided through detailed separation. 

MA-1 was not significantly affected by the presence of Na^+^, Ca^2+^, Fe^3+^, K^+^, or Mg^2+^. In addition, MA-1 was not affected by organic solvents, such as methanol, ethanol, or acetone. However, SDS increased the enzyme activity of MA-1. EDTA and EGTA are metal chelators, and if an enzyme is inhibited by EDTA or EGTA, it is classified as a metalloenzyme [38]. The enzymatic activity of MA-1 was found to be inhibited by EDTA and EGTA. In general, proteases inhibited by PMSF are classified as serine proteases; however, MA-1 did not exhibit any change in enzymatic activity when treated with PMSF. Nonetheless, when MeO-Suc-Arg-Pro-Tyr-pNA, which has substrate specificity for chymotrypsin among the serine proteases, was treated with MA-1, MA-1 effectively degraded MeO-Suc-Arg-Pro-Tyr-pNA. These results suggest that MA-1 is a serine metalloprotease with chymotrypsin-like activity.

MA-1 exhibited fibrino(geno)lytic activity that was approximately five times stronger than that of plasmin. In fact, based on the fibrinogenolytic assay, the Aα, Bβ, and γ-chains were confirmed to be completely degraded within 120 min. Fibrinogen subchain specificity tended to differ for each enzyme. For example, WRL101 derived from *Bacillus* strain has substrate specificity for Bβ-chain [39]. The AprE127 enzyme derived from the *Bacillus* strain had substrate specificity similar to that of MA-1. AprE127 displayed strong Aα-chain decomposition ability and weak Bβ and γ-chain decomposition ability at 80 min [40]. 

Based on the results of the in-tube MA-1 assay, we investigated the efficacy of MA-1 in vivo. First, mice were intravenously (IV) treated with 20,000–80,000 U/kg of MA-1, and blood was collected to confirm anticoagulant activity. MA-1 was confirmed to delay the APTT. In the 80,000 U/kg administration group, APTT delayed coagulation by more than 120 s, and MA-1 was found to be the coagulation factor of the endogenous pathway (I, II, V, X, XII, XI, IX, VIII, prekallikrein, high molecular weight kininogen, and HMWK), and significantly delayed blood coagulation [41].

To confirm the ability of MA-1 to eliminate intravascular thrombosis, an experimental rat model was applied [42]. FeCl_3_ was added to the carotid arteries to induce thrombosis. Nattokinase from *Bacillus subtilis natto* was used as the control because it is a representative fibrinolytic enzyme after various administrations [33]. Nattokinase displayed an antithrombotic effect of 24.0 ± 2.9% when a dose of 80,000 U/kg was administered, while MA-1 displayed an antithrombotic effect of 16.6 ± 1.2% when administered at a dose of 80,000 U/kg. Several mushroom-derived proteases have been reported. Kim et al. purified a thrombolytic protease derived from *Cordyceps militaris* and reported its characteristics; however, its activity was not identified in vivo. Thus, the antithrombotic efficacy of MA-1 could not be considered [43].

Tania et al. compared the antithrombotic effects of four types of mushrooms (*Lentinula edodes*, *Pleurotus ostreatus*, *Pleurotus eryngii*, and *Agrocybe aegerita*) but presented only fibrin(ogen)olytic results [44]. Xiao et al. only revealed the physicochemical characteristics of thrombolytic enzymes, indicating limitations in the potential development of mushroom-derived antithrombotic materials [45].

The effect of MA-1 was analyzed using a thromboembolism-induced mouse model [46]. Thromboembolisms are caused by slow blood flow, excessive blood clotting, and vascular damage [47]. In the present study, blood circulation disorders were induced in mice via the administration of epinephrine and collagen. Nattokinase (80,000 U/kg/day) was also employed as a control. Based on the results, nattokinase exhibited a protective effect of 50%, while MA-1 displayed a protective effect of 40%. There are no reports on the thromboembolism inhibitory effects of mushroom-derived thrombolytic enzymes. Therefore, further studies on the amino acid sequence of MA-1, mushroom-derived proteins, and the development of an oral/injectable thrombosis prevention material based on this analysis are needed in the future. 

To predict the side effects of MA-1, a tail bleeding test was performed using tPA and nattokinase as controls. For antithrombotic agents with rapid effects, delayed coagulation is a disadvantage. As shown in Table 4, tPA caused a coagulation delay of 1600 s. Notably, the clotting times of nattokinase (570 s) and MA-1 (660 s) were more than 1000 s faster than those of tPA. Previously, we reported that C142 (540 s), isolated from *Bacillus* strain C142, had the same coagulation delay time as nattokinase (550 s) [22]. Although MA-1 displayed a longer coagulation delay than C142, the bleeding side effects were expected to be similar to those of tPA.

We isolated a fibrinolytic enzyme fraction (MA-1) from *M. aitchisonii* and investigated its physicochemical properties. MA-1 exhibited excellent fibrinolytic activity compared to the control (plasmin and nattokinase) based on the animal experiments conducted. Although MA-1 is not composed of a single protein, our findings provide a clue to the development of mushroom-derived fibrinolytic enzymes. In future studies, amino acid sequence identification of MA-1 proteins, treatment of thrombotic diseases via different routes of administration (oral, injection), and evaluation of the curative and preventive efficacy are needed.

## 5. Conclusions

Antioxidative protease fraction, MA-1, was purified from *M. aitchisonii*. Both partially purified proteins were found to have similar molecular weights. This enzyme more efficiently degrades fibrin clots via direct fibrino(geno)lysis than it degrades plasmin. MA-1 hydrolyzed the Aα-, Bβ-, and γ-chains of fibrinogen within 2 h. The enzyme may be a chymotrypsin-like metalloprotease based on substrate specificity assay, and the mode of fibrino(geno)lysis. We revealed the ex vivo and in vivo antithrombotic effects of MA-1 via two different mechanisms: anticoagulation activity due to the inhibition of the activation of coagulation factors, and/or anti-clotting activity due to the inhibition of platelet aggregation and blood clot formation. These findings suggest that *M. aitchisonii* containing MA-1 can be a beneficial functional material for preventing cardiovascular diseases and their possible complications.

## Figures and Tables

**Figure 1 antioxidants-12-01558-f001:**
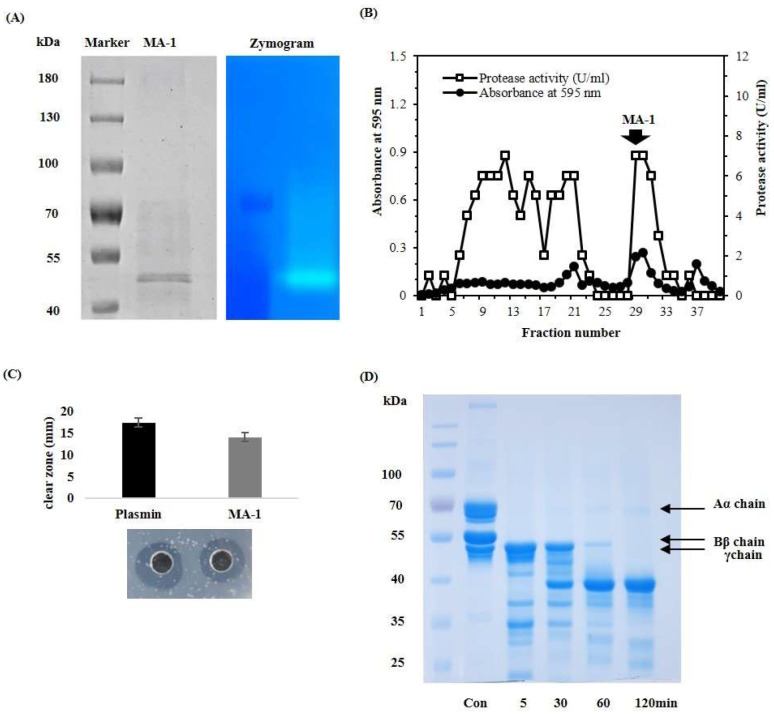
Purification and fibrino(geno)lytic activity of MA-1. (**A**) SDS-PAGE and zymogram of MA-1 in 12% polyacrylamide gel. (**B**) Profiles from Sephadex G-100 column chromatography. (**C**) Fibrinolytic activity on fibrin plate. Fibrin plates composed of 4.5 mg fibrinogen, 0.012 mg agarose, and 0.45 U thrombin in 1 mL of 10 mM Tris-HCl buffer (pH 7.0). Each value represents the mean ± SD of triplicate measurements. (**D**) Fibrinogenolytic pattern analysis of MA-1.

**Figure 2 antioxidants-12-01558-f002:**
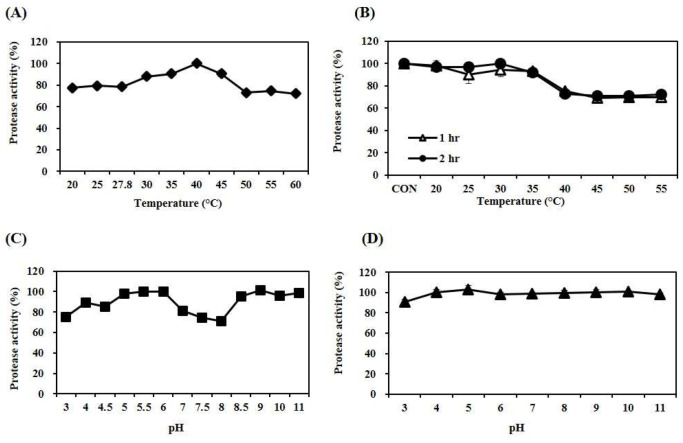
Biochemical properties of the purified enzyme. (**A**) Effect of temperature on protease activity (◆); (**B**) temperature stability (●, △); (**C**) effect of pH on protease activity (■); (**D**) pH stability (▲). Each value represents the mean ± SD of triplicate measurements.

**Figure 3 antioxidants-12-01558-f003:**
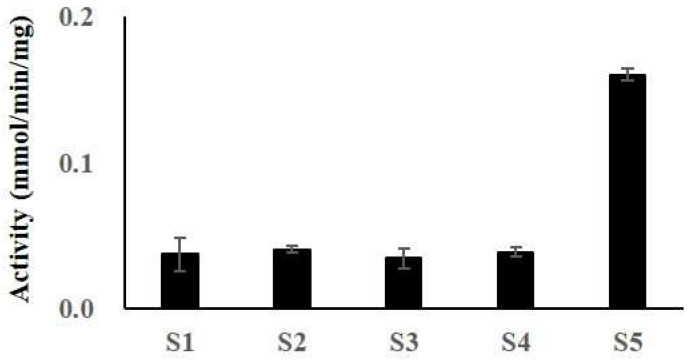
Amidolytic activity of MA-1. The amidolytic activity with several chromogenic substrates was determined by mixing MA-1 and various substrates at 37 °C for 10 min. The amount of *p*-nitroaniline (or substrate) released was determined at 405 nm. Each value represents the mean ± SD of triplicate measurements. S1 (H-D-Ile-Pro-Arg-pNA for t-PA); S2 (H-D-Val-Leu-Lys-pNA for plasmin); S3 (Pyro-Glu-Gly-Arg-pNA for u-PA); S4 (MeO-Suc-Arg-Pro-Tyr-pNA for chymotrypsin); and S5 (H-D-Phe-Pip-Arg-pNA for thrombin).

**Figure 4 antioxidants-12-01558-f004:**
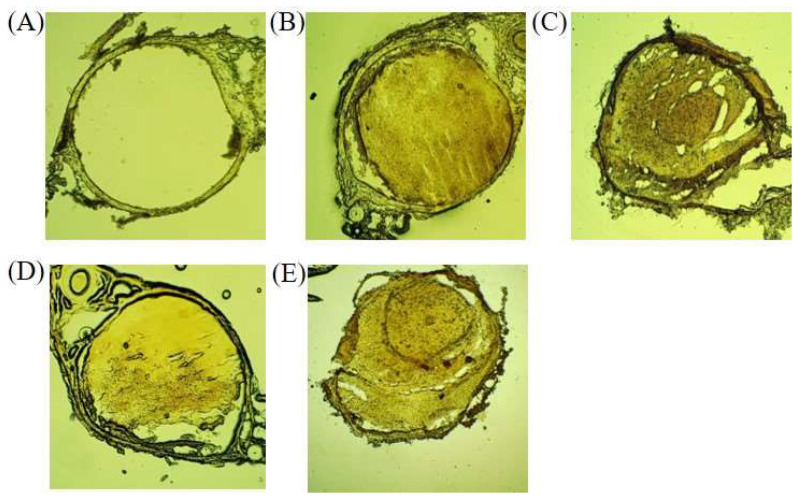
Antithrombotic effect of MA-1. The effects of MA-1 on FeCl_3_-induced carotid arterial thrombus formation in rats were determined. The carotid arteries of rats were occluded with saline (**A**); FeCl_3_ + saline (**B**); FeCl_3_ + nattokinase ((**C**), 20,000 U/kg/day); FeCl_3_ + MA-1 ((**D**,**E**); 10,000 and 20,000 U/kg/day).

**Table 1 antioxidants-12-01558-t001:** Summary of the purification of MA-1.

Step	Total Protein (mg)	Total Activity (U)	Specific Acidity (U/mg)	Fold	Yield (%)
Crude	25	40	1.6	1	100
GPC	0.12	17	141.67	89	0.48

Activity was measured using an azocasein assay. The units of activity were calculated based on a tyrosine standard.

**Table 2 antioxidants-12-01558-t002:** Biochemical properties of MA-1.

	Relative Activity (%)
Control	100 ± 1.9
Metal/inhibitors (1 mM)	
Na^+^	102.6 ± 4.1
Ca^2+^	99.4 ± 1.3
Fe^3+^	95.9 ± 2.3
K^+^	95.1 ± 2.0
Mg^2+^	99.4 ± 3.4
EDTA	75.7 ± 3.2
EGTA	78.0 ± 1.6
PMSF	100.6 ± 2.4
Solvents/surfactants (10%)	
Methanol	98.9 ± 2.2
Ethanol	99.7 ± 1.7
Acetone	100.1 ± 2.0
SDS	116.2 ± 1.6
Glycerol	95.3 ± 2.5
Triton X-100	66.4 ± 2.2
Tween 20	82.4 ± 2.0

**Table 3 antioxidants-12-01558-t003:** Ex vivo anticoagulant activity of MA-1.

	APTT (s)		PT (s)
Control		35.5 ± 1.8	12.3 ± 0.4
MA-1 (U/kg)	20,000	46.1 ± 3.0	12.2 ± 0.4
	40,000	67.0 ± 3.3 *	12.0 ± 0.5
	80,000	120.8 ± 4.8 *	11.5 ± 0.5

Each value represents the mean ± SD of at least 3 independent experiments. One-way ANOVA followed by post hoc Tukey’s test. * *p* < 0.01, compared with the saline-treated control group.

**Table 4 antioxidants-12-01558-t004:** Effect of MA-1 on tail bleeding time.

	Bleeding Time (s)
Vehicle	410 ± 20
t-PA	1620 ± 50 *
Nattokinase	570 ± 20 *
MA-1	660 ± 40 *

Bleeding time was estimated using the tail transection model after intravenous injection of vehicle (saline), t-PA (5000 U/kg), nattokinase (80,000 U/kg), or MA-1 (80,000 FU/kg). Each value represents the mean ± SD (n = 4). One-way ANOVA followed by post hoc Tukey’s test. * *p* < 0.01, compared with the vehicle group.

## Data Availability

Should any raw data files be needed they are available from the corresponding author upon reasonable request.

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
