# Peer review of "Partial Purification and Biochemical Evaluation of Protease Fraction (MA-1) from Mycoleptodonoides aitchisonii and Its Fibrinolytic Effect"

_antioxidants, 2023, doi:10.3390/antiox12081558_

Round 1
Reviewer 1 Report
The manuscript presented by the authors is relevant to the scientific community, the scientific work is well structured. However, it needs some clarifications regarding the results, namely the evaluation of enzymatic activity and the conclusions presented. Some of the graphs presented need correction and some statements in the discussion and conclusions need to be rectified.
2. Material and Methods
2.3 Preparation of the crude extract and purification of MA-1
The authors need to mention the amount of extract applied to the Sephadex G-100 column. In lines 107-109 the authors repeat the same idea.
3. Results and Discussion
In this section the authors only present the results, so the title should be changed to Results.
In Table 1 the authors need to introduce one more column referring to the yield of the purification process and the units of the specific activity should be corrected to U/mg.
In Figure 1 B) the authors should justify why they chose to express the amount/or concentration of protein in Absorbance at 595 instead of mg or mg/ml, the correct way to present it.
In this figure the authors identify the second protein peak as MA1, however almost all fractions from 9 to 21, show proteolytic activity. Fraction 21 shows similar values both at 595 and activity to fractions 29 and 30 identified as the fraction of interest. What are the other fractions and why were they discarded?
Figure 3 The authors need to redraw the figure. What are: Con; S1, S2..S5. In which units are the values shown expressed? The figure should have a specific legend identifying each bar and the specific activity of each synthetic substrate should be expressed in International units (IU) µmol of pNa released/min/mg.
4.Discussion:
The authors state (Page 11 line 423) that Ma-1 is a serine metalloprotease with Chymotrypsin-like activity. The use of azocasein as a substrate in inhibition assays is too general for this type of characterization.
The authors present a synthetic substrate, MeO-Suc-Arg-Pro-Tyr-pNA with a good activity. All biochemical and enzymatic characterization should be performed with this substrate and with more specific inhibitors such as TLCK, TPCK, Phenanthroline, so that it is possible to identify the enzyme. In addition, as we are dealing with a mixture of proteins, proteases of different specificities may be present.
5. Conclusion:
The authors should reword this section. On page 12, lines 487-489 the authors state: "The enzyme may be a chymotrypsin-like metalloprotease based on its sequence homology with other serine or metalloproteases, the effect of protease inhibitors on its activity, and the mode of fi- brino(geno)lysis....".
The authors do not present any amino acid or nucleotide sequence concerning the MA-1 fraction, so we do not know if it is homologous or not!!!! MA-1 shows a reduction of about 25% of the enzymatic activity using a generic substrate and a generic set of protease inhibitors, so it is not possible to make this type of statement.
Author Response
Response to Reviewer’s 1 Comments
Reviewer comment #1
The manuscript presented by the authors is relevant to the scientific community, the scientific work is well structured. However, it needs some clarifications regarding the results, namely the evaluation of enzymatic activity and the conclusions presented. Some of the graphs presented need correction and some statements in the discussion and conclusions need to be rectified.
- Material and Methods
2.3 Preparation of the crude extract and purification of MA-1
The authors need to mention the amount of extract applied to the Sephadex G-100 column. In lines 107-109 the authors repeat the same idea.
Author response #1: Thank you for the appreciation. The amount of protein injected into the column and the volume of the sample are written in red color in section 2.3.
Reviewer comment #2
- Results and Discussion
In this section the authors only present the results, so the title should be changed to Results.
Author response #2: Thank you for your insightful comments. We also agree with your opinion. Among the subheadings of the result section, three subheadings that needed to be specified were changed (3.2 3.4 and 3.5)
Reviewer comment #3
In Table 1 the authors need to introduce one more column referring to the yield of the purification process and the units of the specific activity should be corrected to U/mg.
Author response #3: Thank you for the appreciation. mg/U is mistypo, thus we corrected.
And yield was inserted into table 1.
Reviewer comment #4
In Figure 1 B) the authors should justify why they chose to express the amount/or concentration of protein in Absorbance at 595 instead of mg or mg/ml, the correct way to present it.
Author response #4: Thank you for the appreciation. In general, when using FPLC, the protein elution pattern is identified using the detector. In our study, an open column was used, and fractions showed relative protein elution patterns at 595 nm through the method of bradford assay.
Reviewer comment #5
In this figure the authors identify the second protein peak as MA1, however almost all fractions from 9 to 21, show proteolytic activity. Fraction 21 shows similar values both at 595 and activity to fractions 29 and 30 identified as the fraction of interest. What are the other fractions and why were they discarded?
Author response #5: Thank you for your insightful comments. In addition to MA-1, other active fractions showed lower protein conc than MA-1 and non-specific binding to the membrane when concentrated, so in this study, we focused on MA-1. We plan to study fractions other than MA-1 when mass extraction is carried out in the future.
Reviewer comment #6
Figure 3 The authors need to redraw the figure. What are: Con; S1, S2..S5. In which units are the values shown expressed? The figure should have a specific legend identifying each bar and the specific activity of each synthetic substrate should be expressed in International units (IU) µmol of pNa released/min/mg.
Author response #6: Thank you for the appreciation. The figure was modified by converting the units in Fig. 3 and missing labels were added. Descriptions of S1 ~ S5 have been added to figure.
Reviewer comment #7
4.Discussion:
The authors state (Page 11 line 423) that Ma-1 is a serine metalloprotease with Chymotrypsin-like activity. The use of azocasein as a substrate in inhibition assays is too general for this type of characterization.
The authors present a synthetic substrate, MeO-Suc-Arg-Pro-Tyr-pNA with a good activity. All biochemical and enzymatic characterization should be performed with this substrate and with more specific inhibitors such as TLCK, TPCK, Phenanthroline, so that it is possible to identify the enzyme. In addition, as we are dealing with a mixture of proteins, proteases of different specificities may be present.
Author response #7: Thank you for your insightful comments. As the reviewer is aware, the purification of enzymes is temperature and time dependent. Therefore, we used the azocasein method for rapid detection of proteolytic enzymes. The azocasein method was used for protease detection and purification efficiency.
The study of substrate properties of enzymes is well known for several known substrates. Therefore, we determined what type of enzyme MA-1 was through a substrate-specific reaction test of the partially purified enzyme (Fig3).
We agree that, as pointed out by the reviewer, the effect of several additional substrate and/or inhibitors should be tested. The point we want to emphasize in our study is that it is the first report that partially purified MA-1 is one of the beneficial factors that affect the blood circulation of mushroom (M. aitchisonii). As described in the discussion section, we plan to further subdivide the protein of MA-1 to reveal its function in the future. In particular, once the optimal purification method for MA-1 is established, it is thought that further studies on substrate specificity of pure enzyme should be conducted.
Reviewer comment #8
- Conclusion:
The authors should reword this section. On page 12, lines 487-489 the authors state: "The enzyme may be a chymotrypsin-like metalloprotease based on its sequence homology with other serine or metalloproteases, the effect of protease inhibitors on its activity, and the mode of fi- brino(geno)lysis....".
The authors do not present any amino acid or nucleotide sequence concerning the MA-1 fraction, so we do not know if it is homologous or not!!!! MA-1 shows a reduction of about 25% of the enzymatic activity using a generic substrate and a generic set of protease inhibitors, so it is not possible to make this type of statement.
Author response #8: Thank you for the constructive comments. We deleted and rewritten the parts pointed out by reviewers.
Reviewer 2 Report
In the present article the authors have identified a chymotryptic like metalloprotease activity in the cold water extract of the fungus Mucoleptodonoides aitchisonii. Previous studies have focused in the identification of small molecule bioactive compounds but have not described the analysis of proteins and proteases. Here, the authors have analyzed the proteolytic fraction of the fungus produced by cold water extraction and gel permeation chromatography. The have isolated a fraction that contains two proteins with apparent molecular weight of 48 kDa. These proteins have proteolytic activities and may have potential antithrombotic activities. The study is interesting with potential pharmacological applications.
However, there are several points that need to be addressed.
Specific comments follow.
1. In the abstract it is stated that “The optimum pH, optimum temperature, pH stability, and thermal stability were pH 5.5, 40 °C, pH 5, and 20-35 °C, respectively” but the pH stability is a range, here it is stated 5. Further, the optimum temperature is 40 degrees more than the thermal stability 20-35!!
2. In the protease assay azocasein is digested, this produces a change is the absorbance, that correlates with activity. However, the authors instead of measuring the absorbance they add F-C reagent and measure tyrosine. Where tyrosine comes from?
3. In the methods “The extract (0.1 mL) was mixed with 0.2 M sodium phosphate buffer (0.5 mL), and 152 K3[Fe(CN)6] (0.5 mL, 1% w/v), followed…” What is the pH of the phosphate?
4. How do the authors define the MA-1 enzymatic units?
5. Why do the authors use only male mice?
6. In Table 1 specific activity is U/mg not mg/u
7. In Line 280 PMSF fluoride fluoride is not necessary F stands for fluoride
8. In Line 297 what was the concentration of surfactants?
9. In conclusion it is stated that “The enzyme may be a chymotrypsin-like metalloprotease based on its sequence homology with other serine or metalloproteases” however no such data are provided in the text.
10. Although the analysis of the protein sequence could be the purpose of a new study, it is highly recommended that the authors conduct some more characterization. Since the two bands are very close and it appears in the zymography only one band this could indicate that these two proteins could be a proenzyme and enzyme or a glycosylated form of the enzyme. A deglycosylation reaction is highly recommended. Further incubation for extended periods of time and analysis by SDS-PAGE should be performed to see if there is a change between the intensity of the two bands.
Author Response
Response to Reviewer’s 2 Comments
Reviewer comment #1
In the present article the authors have identified a chymotryptic like metalloprotease activity in the cold water extract of the fungus Mucoleptodonoides aitchisonii. Previous studies have focused in the identification of small molecule bioactive compounds but have not described the analysis of proteins and proteases. Here, the authors have analyzed the proteolytic fraction of the fungus produced by cold water extraction and gel permeation chromatography. The have isolated a fraction that contains two proteins with apparent molecular weight of 48 kDa. These proteins have proteolytic activities and may have potential antithrombotic activities. The study is interesting with potential pharmacological applications.
However, there are several points that need to be addressed.
Specific comments follow.
Author response #1: With due respect, we would like to thank the editor and the reviewer for their review comments. Based on their comments, we have revised our manuscript for resubmission.
Reviewer comment #2
In the abstract it is stated that “The optimum pH, optimum temperature, pH stability, and thermal stability were pH 5.5, 40 °C, pH 5, and 20-35 °C, respectively” but the pH stability is a range, here it is stated 5. Further, the optimum temperature is 40 degrees more than the thermal stability 20-35!!
Author response #2: Thank you for the reviewer's point. The pH stability result part has been modified in red color in the abstract.
Reviewer comment #3
In the protease assay azocasein is digested, this produces a change is the absorbance, that correlates with activity. However, the authors instead of measuring the absorbance they add F-C reagent and measure tyrosine. Where tyrosine comes from?
Author response #3: Protease activity is generally measured based on the amount of tyrosine produced after proteolysis. Therefore, a calibration curve of tyrosine, a degradation product, is required as a standard material to define protease activity. Tyrosine is used for this purpose.
Reviewer comment #4
In the methods “The extract (0.1 mL) was mixed with 0.2 M sodium phosphate buffer (0.5 mL), and 152 K3[Fe(CN)6] (0.5 mL, 1% w/v), followed…” What is the pH of the phosphate?
Author response #4: Thank you for the appreciation. The pH (6.5) of the buffer solution used for measuring the reducing power is added in red color.
Reviewer comment #5
How do the authors define the MA-1 enzymatic units?
Author response #5: Thank you for the appreciation. We defined the unit of proteolytic enzyme as described in section 2.4.One unit of enzyme activity was defined as the ability to produce 1 μg of tyrosine per minute.
Reviewer comment #6
Why do the authors use only male mice?
Author response #6: Thank you for the appreciation. We primarily used male mice for consistent design and operation of the experiment.
Reviewer comment #7
In Table 1 specific activity is U/mg not mg/u
Author response #7: Thank you for the appreciation. mg/U is mistypo, thus we corrected.
Reviewer comment #8
In Line 280 PMSF fluoride fluoride is not necessary F stands for fluoride
Author response #8: Thank you for the appreciation. we deleted the fluoride
Reviewer comment #9
In Line 297 what was the concentration of surfactants?
Author response #9: Thank you for the appreciation. we added the conc of surfactants (10%) in red color
Reviewer comment #10
In conclusion it is stated that “The enzyme may be a chymotrypsin-like metalloprotease based on its sequence homology with other serine or metalloproteases” however no such data are provided in the text.
Although the analysis of the protein sequence could be the purpose of a new study, it is highly recommended that the authors conduct some more characterization. Since the two bands are very close and it appears in the zymography only one band this could indicate that these two proteins could be a proenzyme and enzyme or a glycosylated form of the enzyme. A deglycosylation reaction is highly recommended. Further incubation for extended periods of time and analysis by SDS-PAGE should be performed to see if there is a change between the intensity of the two bands.
Author response #10: Thank you for the constructive comments. We deleted and rewritten the parts pointed out by reviewers.
And we fully agree that, as pointed out by the reviewer, the effect of several additional works should be tested. The point we want to emphasize in our study is that it is the first report that partially purified MA-1 is one of the beneficial factors that affect the blood circulation of mushroom (M. aitchisonii). As described in the discussion section, we plan to further subdivide the protein of MA-1 to reveal its function in the future. In particular, once the optimal purification method for MA-1 is established, it is thought that further studies on substrate specificity of pure enzyme should be conducted.
Round 2
Reviewer 2 Report
The authors have responded to all comments. However a minor comment in the abstract has not been address.
It is stated that "The optimum pH, optimum temperature, pH stability, and 28 thermal stability were pH 5.5, 40 °C, pH 4-11, and 20-35 °C"
If the optimum temperature is 40 then then enzyme should stable to temperature more than 40. Here it is stated 20-35 (even less!!)
After fixing this minor comment the manuscript can be found acceptable. It is not necessary to check it again from my side.
Author Response
Response to Reviewer’s Comments
Reviewer comment #1
The authors have responded to all comments. However a minor comment in the abstract has not been address.
Author response #1: With due respect, we would like to thank the the reviewer for their review comments. Based on their comments, we have revised our manuscript for resubmittion.
Reviewer comment #2
It is stated that "The optimum pH, optimum temperature, pH stability, and 28 thermal stability were pH 5.5, 40 °C, pH 4-11, and 20-35 °C" If the optimum temperature is 40 then then enzyme should stable to temperature more than 40. Here it is stated 20-35 (even less!!)
Author response #2: Thank you for the reviewer's point.
Thanks for the reviewer's comments. In the case of MA-1, the enzyme works at 40C maximal. However, when checking the remaining activity after leaving it at 40C for 1 hour and 2 hours, the activity remained less than 80% compared to the sample stored at low temperature.
However, after 2 hours left at 35C or less, the remaining activity was checked, and more than 80% of the activity remained when compared to the low-temperature stored sample. We believe that the enzyme activity of MA-1 is 40C in a short time, but the durability of the protein is slightly lowered if the storing time continues above 40C.
We confirmed the temperature stability once again to the first author who conducted the experiment. Experimentally, MA-1's optimal temperature (40C) and temperature stability (35C or less) did not match. Therefore, we infer that MA-1 has some structural durability problem, and we estimate that the cause can be identified through accurate sequence analysis in the future.
In addition, the description in the abstract are described in a little more detailed.
The optimum temperature was 40 °C and stability was stable in the range of 20 to 35 °C.
After fixing this minor comment the manuscript can be found acceptable. It is not necessary to check it again from my side.
